# A Systematic Review of Food Allergy: Nanobiosensor and Food Allergen Detection

**DOI:** 10.3390/bios10120194

**Published:** 2020-11-29

**Authors:** Adriano Aquino, Carlos Adam Conte-Junior

**Affiliations:** 1Center for Food Analysis (NAL), Technological Development Support Laboratory (LADETEC), Federal University of Rio de Janeiro (UFRJ), Cidade Universitária, Rio de Janeiro, RJ 21941-598, Brazil; aquinolp@gmail.com; 2Laboratory of Advanced Analysis in Biochemistry and Molecular Biology (LAABBM), Department of Biochemistry, Federal University of Rio de Janeiro (UFRJ), Cidade Universitária, Rio de Janeiro, RJ 21941-909, Brazil; 3Nanotechnology Network, Carlos Chagas Filho Research Support Foundation of the State of Rio de Janeiro (FAPERJ), Rio de Janeiro, RJ 21941-909, Brazil; 4Graduate Program in Veterinary Hygiene (PPGHV), Faculty of Veterinary Medicine, Fluminense Federal University (UFF), Vital Brazil Filho, Niterói, RJ 24230-340, Brazil; 5Graduate Program in Sanitary Surveillance (PPGVS), National Institute of Health Quality Control (INCQS), Oswaldo Cruz Foundation (FIOCRUZ), Rio de Janeiro, RJ 21040-900, Brazil; 6Graduate Program in Chemistry (PGQu), Institute of Chemistry (IQ), Federal University of Rio de Janeiro (UFRJ), Cidade Universitária, Rio de Janeiro, RJ 21941-909, Brazil; 7Graduate Program in Food Science (PPGCAL), Institute of Chemistry (IQ), Federal University of Rio de Janeiro (UFRJ), Cidade Universitária, Rio de Janeiro, RJ 21941-909, Brazil

**Keywords:** biosensing, nanocomposite, Big Eight, PRISMA, food treatment

## Abstract

Several individuals will experience accidental exposure to an allergen. In this sense, the industry has invested in the processes of removing allergenic compounds in food. However, accidental exposure to allergenic proteins can result from allergenic substances not specified on labels. Analysis of allergenic foods is involved in methods based on immunological, genetic, and mass spectrometry. The traditional methods have some limitations, such as high cost. In recent years, biosensor and nanoparticles combined have emerged as sensitive, selective, low-cost, and time-consuming techniques that can replace classic techniques. Nevertheless, each nanomaterial has shown a different potential to specific allergens or classes. This review used Preferred Reporting Items for Systematic Reviews and the Meta-Analysis guidelines (PRISMA) to approach these issues. A total of 104 articles were retrieved from a standardized search on three databases (PubMed, Scopus and Web of Science). The systematic review article is organized by the category of allergen detection and nanoparticle detection. This review addresses the relevant biosensors and nanoparticles as gold, carbon, graphene, quantum dots to allergen protein detection. Among the selected articles it was possible to notice a greater potential application on the allergic proteins Ah, in peanuts and gold nanoparticle-base as a biosensor. We envision that in our review, the association between biosensor and nanoparticles has shown promise in the analysis of allergenic proteins present in different food samples.

## 1. Introduction

Food allergy (FA) is usually described as an immunological reaction. The mechanisms for an allergic reaction involve an Immunoglobulin E (IgE) and non-Immunoglobulin E (non-IgE) mediated responses resulting from exposure to a particular food [1,2,3,4,5]. FA is a fatal condition that can negatively affect well-being, triggering symptoms such as itching, diarrhoea, stomach pain, eczema, shortness of breath, loss of consciousness, and anaphylactic shock [3,6]. FA is a more severe challenge, having become a frenzy in recent years [7]. Some studies have observed the prevalence of food allergies among children, but less into food allergies in adults [8,9]. FA affects more than 1 or 2%, but less than 10% of the population, with a higher incidence in children than in adults [10].

Allergenic compounds have become a concern when it comes to food safety terms. Immunological reactions to allergens differ from food intolerance, pharmacological reactions, and intoxication. According to a report published by the Food and Agriculture Organization (FAO) of the United Nations, the most critical allergenic foods are soybean, cow’s milk, chicken eggs, peanuts, crustaceans, tree nuts, and cereal sources of gluten and fish the called the Big Eight [2,6,9,11,12,13,14,15,16,17,18] (Figure 1). Some foods come with information about the presence and absence of allergenic compounds. Although food labelling is essential to provide consumers with information on the composition of products, accidental exposure to some allergenic compounds can occur. This exposure may occur due to undeclared allergenic substances through food adulteration, cross-contamination, or even fraud [2,10,19].

A preventive strategy can be done by feeding, avoiding foods that contain allergenic compounds, or selecting foods that have undergone some process of industrialization, able to inhibit the properties of allergenic compounds. Industrial processing, such as, for example, pasteurization, can promote change in the allergenic properties of the proteins present in cow’s milk, minimizing or even inhibiting allergenic activity. High temperatures can be responsible for this process [21]. Table 1 shows processes that can be used to reduce allergenic proteins present in food. Another way is to employ a device that can prevent allergic reactions by monitoring food using point of care (POC) biosensors to detect allergens [22].

Detecting allergens in different food matrices is defiant, as allergens can be present at trace levels, and the different compounds present in the various food matrices. Different methods are employed for the analysis, as immunological, genetic, and those based on mass spectrometry, peptidomics and proteomics. The biosensor is a device capable of detecting different compounds such as environmental pollutants [32,33], the vitamin [34], pesticide residue [35,36], and biomolecules in biological [37] or food [38] samples. Biosensors are subdivided into three components, bio-recognition, transduction, and signal processing (Figure 2). Nanotechnology applied to biosensors (nanobiosensors) are considered innovative, sensitive, selective, low cost, and fast techniques and can go through automation and miniaturization and may even replace the classical methodologies [10]. Nanoparticles attribute to nanobiosensors their properties, such as the high ratio of surface area and volume, conductivity, improving performance and furthering its applicability in the detection of allergenic proteins. In this context, this systematic review presents biosensors and nanoparticle (NPs) applications developed to analyze allergens present in foodstuffs (Figure 3) based on the research available in PubMed, Scopus, and Web of Science platforms.

## 2. Methodology

For the selection of the articles used in the systematic review, steps were performed, such as an evaluation of the articles based on their abstracts and titles. The articles were selected, if in abstracts based on the presence of the relationship between allergenic protein/biosensor and nanoparticles. Only publications, original articles, or reviews were selected for the review’s preparation and was restricted only to those published in English. The selected productions were limited to biosensors’ development, the use of nanoparticles and allergenic compounds, and editorials, letters, and doctoral theses were eliminated. Among the selections, articles were included in which the main allergenic foods (the Big Eight), food allergy-symptoms, and causes were considered from the removal of allergenic proteins. The studies related to the development of allergies were considered essential and were then included in the review, as they serve as a knowledge base about the relevance of the study of allergenic proteins, and the impact of these proteins on the individual’s health also regarding the risk of consumption of foods considered allergenic. The dates of delimitation for the selected articles were between 2002–2020. Preferred Reporting Items for Systematic reviews and Meta-Analyses (PRISMA) served as the basis for the selection of papers, as shown in the flow diagram in Figure 4 [39].

### 2.1. Focus Questions

The focus issue was according to the population, intervention, comparison, and outcome (PICO) method. The research questions were based on the following form: (P)-What types of nanomaterials can be used to improve the detection of allergenic proteins? (I)—Which nanoparticle is most used for the detection of allergenic proteins using a biosensor? (C)—What is the best nanobiosensors to the detection of allergenic proteins? (O)—What are the functions of the nanoparticle in the detection of allergenic proteins in food?

### 2.2. Information Sources

Search component 1 (SC1-(P)): Nanoparticle OR Allergen protein OR Big Eight OR Food allergy.

Search component 2 (SC2-(I)): Biosensing Techniques OR Biosensors optical OR Biosensors electrochemical OR Mass-based Biosensors OR Biosensors OR Gold nanoparticles OR Graphene OR Quantum dots OR Monoclonal antibody OR Aptameric biosensor OR Biosensors immune OR Biorecognition OR Nano-Biosensors.

### 2.3. Risk of Bias Assessment

Sources of bias include detection of the same allergenic protein and the absence of results as a limit of detection.

## 3. Results

In the articles, the selection was given priority to biosensors and nanoparticles’ applications to analyze allergenic proteins present in the main allergenic foods. Using the search site PubMed, 407 articles were found (according to the selection criteria) and used, only 22 were considered ideal for the proposal. A total of 125 articles were found for the Web of Science platform and 11 were selected. A total of 176 articles were found for the Scopus platform and 15 were selected. Another 56 articles were added, in addition to those found on the research platforms. Some of these documents were used to fill in the lack of explanations for other articles selected in the databases, such as those that addressed allergenic protein’s clinical aspects.

### 3.1. Animal and Plant Food Allergens

Some allergens from animal and plant sources are proteins [10]. The frequency of allergies is mainly concentrated among the most consumed foods globally: the “Big Eight”. However, the implementation of new food processing and protein sources such as insects, algae, among others, can cause new food allergies [40,41]. The main problems with allergies were foods such as eggs, milk, crustaceans, molluscs, fish, peanuts, tree nuts, soybeans, and wheat. Thus, in sources of allergic compounds include animals and plants, about 160 compounds have been identified [6,42,43]. According to the Food and Agriculture Organization (FAO, 1995) and the Codex Alimentarius Commission (1999), there are different foods that may have allergens which are [40] nuts, almonds, hazelnuts, sesame seeds, buckwheat, oats, celery, mustard, lupins, egg, milk, soybean, wheat peanuts, shellfish, and fish [43].

Eggs are an example of food used as a source of protein of animal origin. Besides, they are important dietary sources of lipids, vitamins, and minerals [44]. However, eggs are the most common source of allergenic compounds found in egg whites and egg yolks, but allergic reactions are more frequently caused by egg white proteins than egg yolk [45,46]. The allergens present in egg whites are Gal d 1: ovomucoid, Gal d 2: ovalbumin, Gal d 3: ovotransferrin, and Gal d 4: lysozyme. Allergens such as Gal d 5: α-livetine and Gal d 6: the precursor to vitellogenin-1 is found in egg yolk [43]. Among these proteins, ovalbumin is the most abundant protein in an egg that more induces allergic reactions in individuals [47].

Milk was the most frequently reported by children as allergic to 38.5% and was the second product among adults [21]. Cow’s milk has about 20 allergenic proteins. Among them are α-lactalbumin (Bos d 4), β-lactoglobulin (Bos d 5), BSA (Bos d 6), bovine immunoglobulins (Bos d 7), and casein allergens (Bos d 8) [21,43,48,49]. In adult life, the most common allergic diseases are caused by hypersensitivity to oilseeds, fish meat, mollusks, and fruit [40,50].

Allergies to seafood, such as shrimp, affect approximately 6.5 million people in one year. Even in cooked seafood, some allergenic proteins resist the heating process. An example is tropomyosin (TROP), a protein present in shrimp. Shrimp allergens are calcium-binding protein, arginine kinase, light chain myosin, isomerase triosephosphate, sarcoplasm, TROP, and troponin [17,43,51,52,53].

Peanuts are an example of allergenic foods of plant origin, and the number of cases of peanut allergies seems to have increased suddenly in recent years [54,55]. Currently, 12 allergenic compounds are identified in peanuts. Some of these are identified as arachnids Ara h 1 to Ara h 13. Arachin h 4 has been renamed Ara h 3.02. Ara h 8 is one of the major peanut allergens Ara h 8. Other allergens include Ara h 1, 2, 3, and 9. Moreover, Ara h 6 is an allergen, identified in peanuts, and has characteristics such as resistance to digestion and heat. Among them, Ara h 1 is the most abundant peanut allergen [15,16,43,56,57].

Another plant-based food allergen, soybean, is among the “Big Eight” most allergenic foods and is widely used in food products. The only solution to prevent allergenic reactions is to avoid foods containing some allergenic compounds. Since soybeans are ubiquitous in vegetarian products, avoiding them has become increasingly difficult. Soy provides many health benefits, so it has become necessary to eliminate the allergenic components present in soy or leave it within a safe limit because soy is widely used in processed foods. However, processed foods that do not contain soybeans are challenging to access. The elimination of their allergenic compounds is practically impossible, making it necessary to investigate soy ingredients and products. So it is essential to perform what gives the allergic capacity of soybean-based ingredients and products [58].

Cereals are vital sources of vitamins, fiber, minerals, and proteins. However, wheat is as important a crop as it is an allergenic food [59]. In food products such as cereals, cookies, bread, drinks, and other products, gluten can be found. Gluten has more than 50 proteins, which are divided into two groups: glutenins and gliadins. Gluten is allergenic, which can be present in the insoluble fraction, which is: Tri a 19: ω5-gliadin, Tri a 20: γ-gliadin, Tri a 21: γ/β-gliadin, in the group of gliadin and Tri a 26: high molecular weight gluten and Tri to 36: low molecular weight gluten, in the glutenin group. The group of glutenins are covalently bound proteins, while gliadins are covalently unbound and are soluble in alcohol. Allergens in fruits are present in fresh and dried fruits and can pose a threat since they can be present in some processed fruit-based foods. Some allergens are Jug n 2, Jug r 2 and Jug r 4 (termites), Jug n 1, Jug r 1 and Jug r 3 (prolamines) and Jug r 5 (profilins) [43]. More allergenic proteins and their classifications in different foods can be found in the WHO/IUIS allergen database (http://www.allergen.org).

### 3.2. Allergen Detection

With many allergen foods, adulteration, contamination, and sustainable novel food protein sources becoming necessary, applying analytical techniques that are cheap, fast, sensitive, and accurate along the production line has also become necessary [2,4,10]. Counterfeiting food products can have irreversible health consequences, as their consumption can trigger sensitized individuals. In this case, evaluating the authenticity of the product is very important to identify and assess its value and to avoid unequal competitiveness and to protect consumers against the fraud that is common in the food industry. An alternative to estimate the authenticity of foods is the enzyme-linked immunosorbent assay (ELISA), which has become the most used methodology in immunological methods for the determination of allergenic protein [3,58,60,61,62,63,64]. However, the ELISA method has disadvantages, as it has a complicated succession of steps, a long analysis time and expensive reagents, and during the analysis, cross-reactions can occur [10]. ELISA and mass spectrometry (MS) are tools that can determine the proteins present in different foods. Furthermore, tools that are based on the detection of nucleotide sequences such as DNA or RNA of allergenic proteins are also used, such as polymerase chain reaction (PCR) and its variants such as multiplex-real-time PCR, PCR-ELISA, and duplex PCR [10,65,66,67] (Table 2). In addition, biosensors have been developed as a viable alternative for the detection of allergens, and have advantages such as sensitivity, selectivity, low cost, and the possibility to miniaturize the device [9,68,69].

### 3.3. Biosensor Technology

The technological development of biosensors has become important in several fields, such as biomedicine, drug discovery, food safety, and environmental monitoring. The analyses carried out using biosensors are called detection by labels (detection of targets or marked) and without labels (targets that are not marked) [80]. A biosensor has an integrated receptor and a transducer device that converts the biological recognition element in a measurable chemical-physical output signal proportional to the concentration of the target analyte. When the bioreceptors selectively identify or interact with the analyte, a signal will be produced that can be in the form of a mass change, heat, charge, and pH. In the biosensor, bio-recognition, or bioreceptor, it recognizes target analytes, generating the signal that is transferred to the transducer that performs the detection. The bioreceptors most commonly used in biosensors are enzymes, cells, antibodies, molecularly imprinting polymer-based molecular recognition, and nucleic acids. In this way, the methods for allergen analysis can be mainly divided into immunological assays and those that are DNA-based. Immuno-biosensors are based on the specific binding between a known allergen or a target (antibody) and the DNA-based (genosensor) nucleotide sequence [10,14,81].

Antibodies that are monoclonal (MAb) or polyclonal (PAb) can be both used to develop immunosensors. PAb connects different epitopes, and the MAb is very specific for a single epitope of a particular allergen. MAb has advantages in comparison to PAb, due to the specific binding to the target protein, namely decreasing in non-specific binding or reduction in cross-reactions with different proteins present in food allowing the development of more sensitive and specific immunosensors. In immunosensors, allergenics, proteins, or antibodies are immobilized on the surface, and changes in specific optical properties accompanied by antigen–antibody reactions on surfaces can be measured through optical transducers such as surface plasmon resonance (SPR), localized SPR (LSPR), imaging SPR (iSPR), or resonance enhanced absorption (REA). The light source and wavelength—polychromatic or monochromatic—are essential in the SPR sensor development. In addition to these, other transducers can be used, such as electrochemical, impedance, piezoelectric/electrochemical, or voltammetry. Further details about these types of transducers can be found in some literature reviews [10,82,83,84].

The development of an antibody-based biosensor can be an opportunity to reduce the required amount of antibody used. The immunosensor is reusable, that is, in biosensors, the antibodies can be used several times [61,85]. Biosensors present an excellent potential for monitoring allergens due to their high sensitivity and selectivity, low cost, and rapid analysis [68]. The characteristics of the antigen–antibody binding interactions make antibodies and monoclonal antibodies the ideal alternatives for protein detection in allergenic biosensors. The main trends in applying antibodies in the biosensor development process include the regeneration of antibodies, the development of conjugated enzymes, and antibody immobilization methods [43]. However, immune recognition methods present a disadvantage in their short shelf-lives [10]. The connection between an antigen and an antibody involves Van der Waals interactions, hydrogen, and electrostatic bonds, as well as the hydrophobic forces between the antibody and an antigen region, the epitopes [10]. An interesting approach was performed by Alves et al. (2015) that investigated the applicability of a mouse monoclonal anti-Ara h 6 IgG1 (clone 3B8 B5) antibody, and a biotinylated monoclonal anti-Ara h 6 IgG1 (clone 3E12 C4 B3) antibody to detect Ara h 6 in peanuts in foodstuffs using voltametric biosensing [77].

DNA-based biosensors (genosensor) are devices in which a biological recognition element is an oligonucleotide sequence. These sequences can recognize a complementary sequence target (RNA or DNA) by a hybridization reaction. The DNA-based biosensor is very specific and sensitive. Besides, allergen encoding-DNA levels are not always correlated with the allergen protein’s presence, the same as when foods are fortified with purified protein. DNA or aptamer-based sensing and other types of transducers can be used with this aim. With a genosensor, it is challenging to compare the limit of detection obtained for proteins by DNA evaluation, but the genosensor is highly sensitive. Another type of nucleotide-based biological recognition element is aptamers, single-stranded DNA or RNA oligonucleotides, with a specific sequence that holds a high affinity towards a target molecule [10]. The bioreceptors and aptamers (aptasensors) have advantages over antibodies, such as eliminating batch to batch variability, low cost, and thermal stability. Aptamers are selected from synthetic nucleic acid libraries using the iterative SELEX (Systematic Evolution of Ligands by Exponential enrichment) for different targets such as DNA, RNA, or peptide libraries [86,87]. They are among the biological recognition elements (immunosensors) that have been used more in the last few years, followed by genosensors and aptasensors. Advances in materials science has made it possible to combine novel selective bio and biomimetic receptors with nanoparticles (Figure 5) that contribute to the optical and electrochemical properties that allowed the development of novel nano-biosensors that can be used in food safety. The advantages of the nano-biosensor are in its detection of allergenic proteins with low limit of detection, by using small volumes of reagents, reduced analysis time, or even detecting multi-analytes, among other possibilities [9].

### 3.4. Nanotechnology in Allergenic Detection R

Nanotechnology has become, in the last two decades, a highly interdisciplinary area that is rapidly converging electronics and surface sciences and areas of life sciences, which encompasses chemistry, food chemistry, biotechnology or materials science, biochemistry, and biology [88,89]. By definition, nanotechnology is understood as the science that uses materials on the nanoscale with sizes ranging from 1 nm to 100 nm and that offer the opportunity to answer questions related to food safety, such as, for example, the detection of allergenic proteins. Nanotechnologies can significantly benefit industries and establish a high impact on the food industry, covering all processing, including food security [90]. The use of nanotechnologies in allergen detection mainly involves nanoparticles, with gold nanoparticles and quantum dots (QDs) being the most used. However, novel nanoparticles are currently being developed and applied due to these characteristics, which allow for faster, low-cost, sensitive, and specific detection [68,88].

Nanotechnology is also promising for detecting different substances present in food products and biosensors, which can be based on nanoparticles and can play a significant role. In combination with nanoparticles for food safety, the application of biosensors has been favored by advances in nanoparticles, since combining biological components with physical-chemical detectors has enormous benefits for the detection and analysis of food allergens. Compared to traditional methods, biosensors based on nanoparticle have some crucial advantages, including low cost and a shorter analysis time [90,91]. Nanoparticles with more applied approaches to food allergen detection are metal oxides such as silver, gold, titanium, carbon and graphene, and quantum dots (Table 3). Novel nanomaterials are under development, with an emphasis on magnetic nanomaterials [43,68]. In addition to associations of nanomaterials such as gold nanoparticles, multi-wall carbon nanotubes- poly-ethyleneimine (AuNPs-PEI-MWCNTs), QDs-aptamer-GO, or different employed materials for different functions such as the application of magnetic nanoparticles for the concentration of target compounds, there has followed the detection of additional functionalized nanoparticles, for example, GO-mAb and HRP-Ab-AuNPs [42,92].

### 3.5. Immobilization Strategies and Nanoparticles-Based Sensors

Biosensors are prepared by modifying the surfaces like metal or carbon using biomaterials [80]. In this sense, electrochemical or optical detection, for example, can be better, modifying nanoparticles surfaces with component bio-recognition. The emergence of nanoparticles in association with biosensors has allowed the detection limit to be reduced as well as the time of analysis and the volume of samples compared to classic methods and materials [88]. Several types of nanoparticles, with different materials such as silver, superparamagnetic iron oxide nanoparticles, gold (AuNPs), carbon nanotubes (CNTs), metal-organic frameworks (MOFs), quantum dots (QDs), graphite, and grapheme can be synthesized as well as immobilized proteins and DNA (RNA) [80,95,96].

#### 3.5.1. Immobilization Strategies and Nanoparticles

An essential step in biosensors’ development is the immobilization of the biorecognition element in the substrate’s surface. Different methods of immobilizing biological recognition compounds on different materials’ surfaces can be used, whether nanostructured or not. Biological materials can be immobilized directly on the surface or indirectly using intermediate ligands.

The combination of biological materials and nanoparticles are promising in the manufacture of biosensors, due to the characteristics of nanostructured materials such as high and effective surface, which promotes a more significant interaction between biorecognition elements and the substrates of the materials, improving the detection process, and consequently the analytical performance of the sensor. However, parameters have to be optimized, as different variables can influence the immobilization efficiency, such as protein concentration, structure, surface descriptors, pH, and ionic strength. These involve factors like hydrophobicity, charge, topography, and chemical groups [97]. The main forces present in non-covalent immobilizations occur between donor-acceptors from the SH groups of molecules (proteins or DNA). In hydrophobic interactions, they also occur in immobilizations where amino acids such as tryptophan are present, and Coulombs between NH2 groups of amino acids such as lysine and citrate ions on the surfaces of gold nanoparticles (AuNPs) [98].

The immobilization of biomaterials on the surface, whether in the transductor or nanoparticle, is considered the most critical step in developing a biosensor’s development. In the biosensor, the immobilized biomolecule has to keep active for the biosensor to have functionality. During the immobilization process, active sites are responsible for recognition and must not be harmed. In the process of choosing an immobilization method, it is necessary to focus on essential factors, such as the type of sample, signal transduction, and the possibility of biosensor reuse [99].

The strategies for immobilizing biomolecules on the material surface are: directly on the material surface (Adsorption), avidin-biotin system, and self-assembled monolayers (SAMs). Adsorption is the simplest method, however, and the least reliable because, during the process, the biological material is randomly fixed on the substrate, which may affect the performance of the biosensor [92,99]. Avidin-biotin is a simple and effective system of anchoring biomolecules to a surface. The interaction of this system is non-covalent, which attributes high stability; however, it is a system with a high cost [99] and SAM systems are mainly obtained by immersing a gold plate in a solution of a surfactant diluted in a high-purity solvent, the packaging and thickness of the monolayer are formed from the radical attached to the sulfide, for example, the groups of alkanethiols forming self-organized monolayers and then, the bio-molecule is linked to the other end of the thiol [99].

Because it has a large surface area, nanoparticles became important in immobilizing enzyme adsorption, as it attributes better stability to catalytic activity. Moreover, some nanoparticles can improve detection systems, whether optical, electrical, chemical, or magnetic. However, immobilization methods have advantages and disadvantages. For example, for the physical adsorption of antibodies, it is necessary to avoid using reagents that cause protein denaturation. The advantage is the selectivity attributed from biological materials to sensors [97].

#### 3.5.2. Nanomaterials Based on Gold (AuNPs)

AuNPs are widely used due to the large surface area, conductivity, compatibility with biological materials, and the broad application of the conjugates of AuNPs are due to the optical and electromagnetic properties [92,98]. Phenomena differences are observed using gold nanoparticles; a recent operation has been reported on how to obtain the surface intensified Raman scattering substrate (SERS) from the localized surface plasmon resonance field (LSPR), which is an optical phenomenon generated by a wave of light trapped in NPs, resulting from the interactions between the incident light and surface electrons [100].

Noble functional gold and silver NPs have been widely used for the colorimetric detection of different compounds. Gold nanoparticles are used because they are stable, easy to synthesize, compatible with different bio-affinity agents, and can be easily detected [43,56]. Gold nanoparticles are widely used in combination with other nanoparticles to improve the detection system; for example, gold particles combined with quartz crystal microbalances provide excellent results when combined with biosensors [10].

Tran and collaborators (2018) developed an aptamer-based biosensor for food allergen determination using graphene oxide/gold nanocomposite on a paper device with hydrophobic channels, set by a wax printer on filter paper. Functionalized gold nanoparticles (AuNP) were used to identify allergens arachin (Ara h 1) in peanuts, β-lactoglobulin (β LG) in milk, and tropomyosin (Pen a 1) for shrimp and other shellfish presence by colorimetric detection [89]. Gold nanoparticles were used to determine, for example, that peanuts have Ara h 1. One of the main allergens present in peanuts is a 7S globulin that has a similar structure to that of vicillin and that can be recognized by IgE antibodies and can be used as a marker to identify the presence of peanuts in food. A biosensor using nanomaterials was developed with the ability to chemically detect Ara h 1 using gold coating and electrodes printed on the carbon screen [43,101].

Sun et al. employed modified gold nanoparticle-polyethyleneimine-multiwalled carbon nanocomposite nanotubes (AuNPs-PEI-MWCNTs) in a label-free voltametric immunosensor system for the detection of kidney bean lectin (KBL). For this purpose, a polyclonal antibody was immobilized on the electrode modified with recombinant staphylococcal protein A, and obtained a good linear response (R^2^ = 0.978) to KBL with a detection limit of 0.023 μg/mL [92].

#### 3.5.3. Nanomaterials Based on Graphene Oxide (GO)

Nanomaterials based on graphene oxide (GO) have received significant attention in numerous applications with biosensors due to their optical, electronic and thermal biocompatibility properties. GO can be applied as an efficient suppressor for various fluorophores due to the transfer of non-radioactive electronic excitation energy between the fluorophore and GO in addition to its broad absorption cross-section, which provides high efficiency. GOs can interact with various molecules such as amino acids, peptides, and proteins, enabling fluorescence detection like gold nanoparticles. With its fluorescence adsorption capability, GO has been increasingly applied in the manufacture of Förster Resonance Energy Transfer (FRET) biosensors. Biosensors manufactured with fluorescent material have excellent chemical stability and efficient and stable signals compared to traditional organic dyes [68]. Zhang et al., in 2016, employed an aptamer-based assay immobilized on the surface of graphene oxide (GO) and fluorescence for the detection of tropomyosin; with this system, a limit of detection of 4.2 nM was possible and worked in the 0.5 to 50 μg·mL^−1^ concentration range [94].

#### 3.5.4. Nanomaterials Based on Magnetic Particles (MPs)

Magnetic particles (MPs) are considered promising nanoparticles for the development of biosensors. The intrinsic magnetic properties of magnetic particles, such as magnetoresistance and giant magnetoresistance, can act as versatile magnetic labels that can be used to indicate molecular interactions [81]. The magnetic properties of the MPs make it possible to capture the targets, consequently enriching and amplifying the readout strategies, attributing a low limit of detection to the biosensors [102].

The magnetic particles (MPs) that have the magnetic core and a non-magnetic shell use bioassays; the outer layer has to be chemically activated to allow the attachment or adsorption of biomolecules [103]. MPs belonging to the group of nanomaterials bring, as a great advantage, easy manipulation through an external magnetic field. MPs allow the combination of modifying the surface of the nanomaterial with a specific capture agent, such as, for example, antibodies, DNA, or glycan, for the selective binding of the molecule of interest [88]. The parvalbumin is an allergenic protein that can be detected by a magnetic nanoparticle probe with antibody-binding immunoassays and side flow [43].

Speroni et al. (2010) developed an Enzyme-linked immunosorbent assay (ELISA) based on antibody-coated magneticmicro-particles (MPs) for the detection of Ara h 3/4 allergen in food. ProteinA-Pn-b and MP-NH2-PAMAMG 1.5-Pn-b immunosupports, were tested. The limit of detection was found to be 0.2 mg with the linear response range from 2.5 to 15 mg peanuts/kg [104].

#### 3.5.5. Nanomaterials Based on Quantum Dots (QDs)

Quantum dots (QDs) are another nanomaterial that can be applied to the detection of allergenic proteins and in biosensors like fluorescent material they have excellent chemical stability, efficient and stable fluorescent signals, and superior biological probes compared to traditional organic dyes [68]. The optical detection system is generally based on the fluorescent properties of quantum dots (QDs) [88]. Weng and Neethirajan (2016) developed a microfluidic system integrated with quantum dot (QDs) nanoparticles and aptamer functionalized graphene oxide (GO) for detection (Ara h 1). It has a detection limit of 56 ng/mL [68].

The application of nanoparticles in the determination of allergenic proteins in foods, in general, has brought as benefits the ability to obtain the limit of the detection, as shown in Table 3. Thus, it was observed that the analytical methods that use AuNPs were the ones with the best limit of detection [59].

## 4. Conclusions and Perspectives

Allergic patients can be exposed to allergenic proteins by consuming products. In this sense, the industry has taken measures to remove or reduce allergenic compounds in processed foods. In some industrialized products, allergenic substances can be not indicated on labels, due to fraud or uncontrolled cross-contamination, making the development and application of strategies to evaluate food allergens in several products necessary. In this sense, devices such as biosensors associated with nanoparticles have been developed for the determination of allergenic compounds. Among nanomaterials, gold nanoparticles are the most used, due to their ease of synthesis. Besides, research focuses on the synthesis of novel nanoparticles for application in analyzing allergenic compounds in food and associating nanoparticles improving detection. In addition to considering the “Big Eight”, which trigger allergic reactions in both adults and children, in this systematic review, we established a bridge among food science and nanotechnology and biosensors. Furthermore, there is still a need for researchers to search for novel sources of allergenic food and allergenic compounds that are generated after processing allergenic foods or introducing other sources of protein such as insects, among others, and this leads to the need to develop methods of detection in addition to conventional ones, such as the use of biosensors with nanoparticles.

Among the nanoparticles most used for the immobilization of biological materials is AuNPs due to their large specific surface area, high surface free energy, good conductivity, and biocompatibility. Associated with this, we can consider as a perspective the use of different component transduction systems together, such as optoelectrical systems and the use of nan-sized molecularly imprinted polymers such as artificial antibodies and hydrogel nanoparticles for the detection of allergenic proteins in foods. The significant trend observed among the new publications include alternative sources of proteins, processing of allergenic foods and the associations of different transducer systems and nanomaterials in association with novel absorbents (immobilization methods) to improve the detection limits of biosensors.

## Figures and Tables

**Figure 1 biosensors-10-00194-f001:**
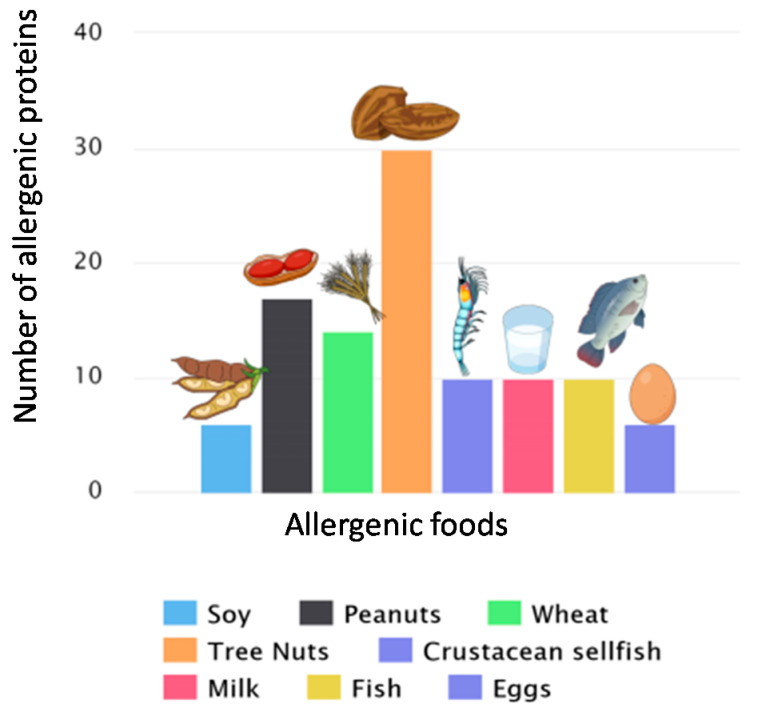
Scheme of the number of allergenic proteins present in the main allergenic foods. This scheme separated between the main plant and animal sources. In this one, the “Big eight” represented the main allergenic foods [20].

**Figure 2 biosensors-10-00194-f002:**
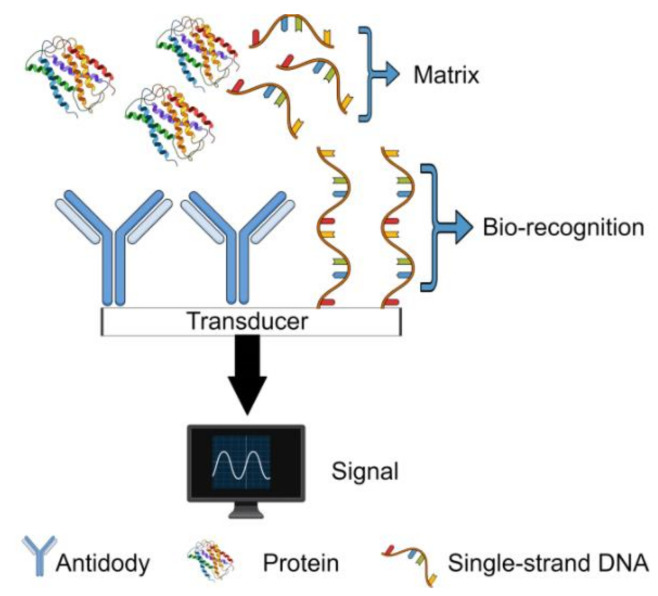
Schematic diagram of the components of a biosensor.

**Figure 3 biosensors-10-00194-f003:**
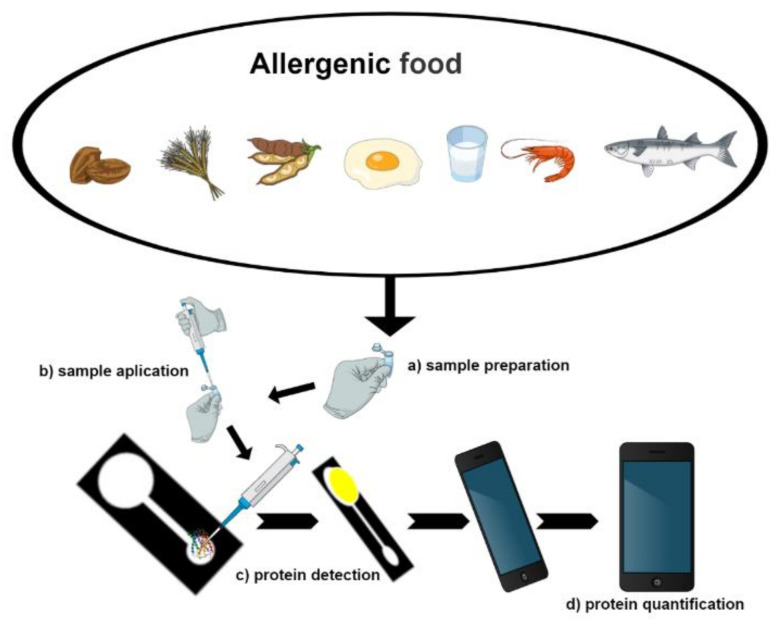
Scheme of the application of different types of food matrices for the analysis of allergenic compounds using biosensors. Analysis of allergenic compounds involves the following steps: (**a**) extraction of allergenic compounds from food and sample preparation steps as necessary; (**b**) application of samples on the biosensor; (**c**) capturing of images with a camera, followed by detection; and (**d**) quantification of the proteins present in each study sample.

**Figure 4 biosensors-10-00194-f004:**
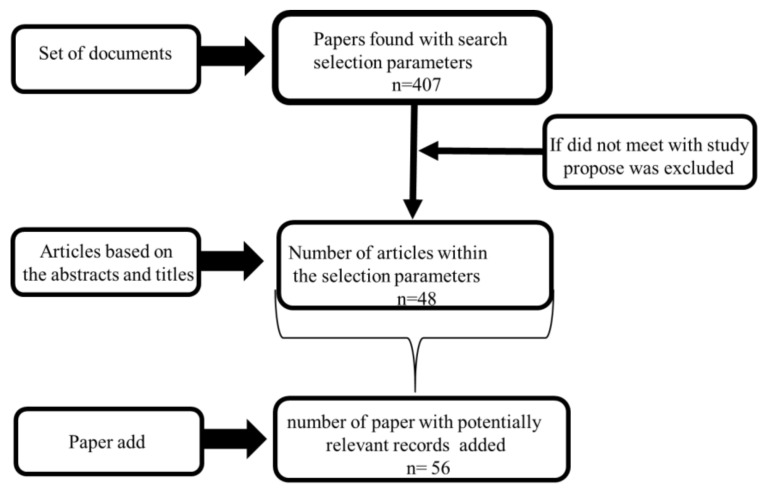
Paper selection scheme.

**Figure 5 biosensors-10-00194-f005:**
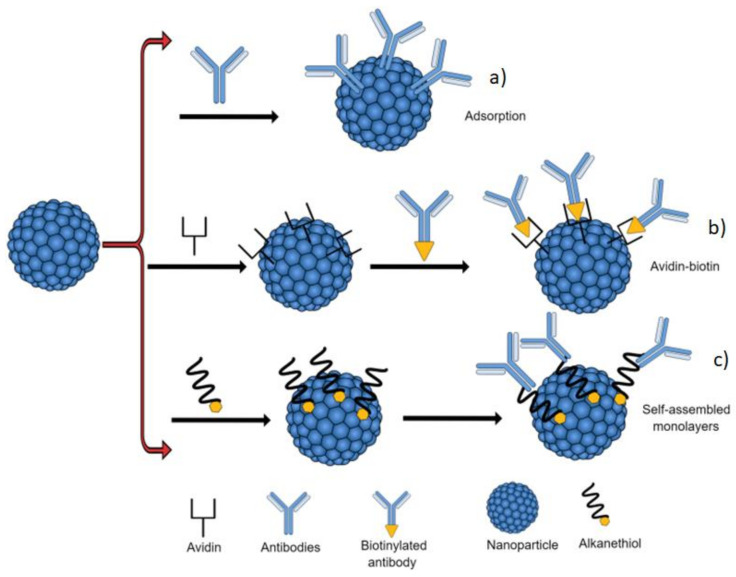
Schematic representation of the antibody immobilization strategies. (**a)** adsorption, (**b**) avidin-biotin, and (**c**) self-assembled monolayers.

**Table 1 biosensors-10-00194-t001:** Processes used to remove or reduce the presence of allergenic compounds in foods, which can be used by the food industry.

Treatments	Allergenic	Matrix	References
Genetic engineering	Profilin	Tomato	[23]
Genetic engineering	Ara h 2	Peanut	[24]
Genetic Modification	Gly m Bd 30 K	Soybean	[25]
Heat Treatment	Lactoglobulin, Lactalbumin, Ovalbumin	Milk, Eggs	[26]
Enzymatic Treatment	Ara h 1 and Ara h 2	Peanut Kernels	[27]
Enzymatic Digestion and Ultrafiltration	β-Lactoglobulin	Cow’s Milk	[28]
Microwave and Enzymatic	β-Lactoglobulin	Cow’s Milk	[29]
Irradiation	β-Lactoglobulin, Albumin, and Tropomyosin	Shrimp, Egg, Milk	[30]
High Hydrostatic Pressure	16 kDa Albumin, R-Globulin, and 33 kDa Globulin	Rice	[31]

**Table 2 biosensors-10-00194-t002:** Methods used for the detection of different allergenic compounds present in food matrix.

Matrix Animal	Allergenic	Method	References
Eggs	Ovalbumin	ELISA	[70]
Milk	Bovine k-casein	Biosensor	[71]
Fish	Cod and pollock	Real-Time PCR	[72]
Fish	Parvalbumin	Immunoblot	[73]
Crustacean Shellfis	Tropomyosin (octopus *Octopus vulgaris*)	ELISA	[74]
Matrix Plant			
Tree nuts	Cor a 9 (hazelnut)	ELISA	[75]
Wheat	Gliadin	Emmunomagnetic beads	[76]
Peanut	Ara h 6	Voltammetric biosensing	[77]
Peanut	Arah	Real-Time PCR Multiplex	[78]
Soybeans	Beta-conglycinin	ELISA	[79]

**Table 3 biosensors-10-00194-t003:** Nanoparticles applied in systems for the detection of different allergenic compounds.

Nanoparticle	Allergenic	Matrix	Caracteristics (Detection Limit)	Sensor Type	References
Gold nanoparticle	Ara h 6; Gly m Bd 28K; and 2S albumin	peanut, soybean and sesame	<0.5 nM	Genosensors	[3]
Carbon	Gluten	food samples based in cereals grains	0.54 mg L^−1^	Immunosensor	[93]
Graphene oxide	tropomyosin	shrimp	0.15 μg·mL^−1^	Immunosensor	[94]
MIP Nanoparticles	α-Casein	food manufacturing environments	0.127 ppm	Immunosensor	[1]
Gold nanoparticles-polyethyleneimine-multiwalled carbon nanotubes nanocomposite	kidney bean lectin (KBL)	kidney beans	0.023 μg/mL	Immunosensor	[92]
GO-AuNP complexes	Ara h 1; β-lactoglobulin (β LG); tropomyosin (Pen a 1)	peanuts; milk; shrimp	7.8 nM (Peanuts), 12.4 nM (Milk), 6.2 nM (shrimp)	Genosensors	[89]
Goldnanoparticle-modified SPCE	Ara h 6	peanut	0.27 ng/mL	Immunosensor	[77]
Quartz crystal microbalance (QCM) chip and gold	gliadin	wheat, barley, oat, rice, foxtail millet, corn, buckwheat, and soybean were	8 ppb	Immunosensor	[59]
Multilayer graphene–gold nanocomposite	Ara h 1	peanut	0.041 fM	Genosensors	[38]
Quantum dots (QDs) aptamer functionalized graphene oxide (GO)QDs-aptamer-GO complexes as	Ara h 1	peanut	56 ng/mL	Genosensors	[68]
L-cysteine/gold nanoparticle (AuNPsCys)-modified	tropomyosin	shrimp	0.15 μg·mL^−1^	Immunosensor	[17]
Graphene oxide (GO) and gold	Parvalbumin	Fish	4.29 ng/mL	Immunosensor	[42]

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
