# Peer review of "A Systematic Review of Food Allergy: Nanobiosensor and Food Allergen Detection"

_biosensors, 2020, doi:10.3390/bios10120194_

Round 1

Reviewer 1 Report

In this manuscript, the association between biosensors and nanoparticles in the analysis of allergenic proteins was reported. The work is something novelty, however, it was not well organized and there are still some issues needs to be addressed. It can be acceptted after “Major revision”.

  1. The application references of Several types of nanoparticles in this article is not thoroughly. Please make sure it is sufficient.
  2. The novelty of the manuscript is not thoroughly presented in the abstract and introduction parts.
  3. The subject of the article is biosensors. Please check whether the reference No. 70 discussed in the nanomaterials of magnetic particles (MPs) fits the subject?
  4. Other food safety problem should also be introduced in the introduction part, such as the pollution, the food additives and the pesticide residue, though the main concerns are allergenic proteins. It is of great importance to the whole manuscript.
  5. What’s the advantage and functions of using nanoparticles in detecting allergenic proteins? There is necessarity to state the main role of nanoparticles and the relationship of them in a separate part.
  6. Is it possible to point out the future research direction of preparing biosensors to detect allergic proteins?
  7. The organizational structure of the article is a bit of chaotic, and the thinking is not clear enough. It needs to be well discussed for more clear. The classification should base on the methods or the objects or the nanomaterials. After selecting a category, other examples and narratives are interspersed, rather than listing various classifications and repeated station.
  8. Some refs about food samples detection should be added, such as 10.1016/j.foodchem.2020.127785, 10.1016/j.foodchem.2019.04.034,10.1016/j.foodchem.2020.127987, 10.1016/j.aca.2014.10.016, 10.1016/j.foodchem.2020.127974,10.1021/jf052489r.

Author Response

1. The application references of Several types of nanoparticles in this article is not thoroughly. Please make sure it is sufficient.
Response: Thanks for your consideration. We added more references for nanoparticles as suggested by the reviewer. References 80; 24; 59; 23; 22 and 21.

2. The novelty of the manuscript is not thoroughly presented in the abstract and introduction parts.
Response: Thanks for your observation and sorry for our mistake. According to the reviewer’s suggestion, more sentences were added.
- Nevertheless, each nanomaterial has showed different potential to specific allergen or class (lines 47-48).
- Among the selected articles, it was possible to notice a greater number of studies based on the allergic proteins Ah, in peanuts and gold nanoparticle-based as a biosensor (lines 54-56).

3. The subject of the article is biosensors. Please check whether the reference No. 70 discussed in the nanomaterials of magnetic particles (MPs) fits the subject?
Response: Thanks for your observation. We add one more sentence to explain the addition of other nanoparticles. Gold nanoparticles are widely used in combination with other nanoparticles to improve the detection system, for example. (Lines 433, 434).

4. Other food safety problem should also be introduced in the introduction part, such as the pollution, the food additives and the pesticide residue, though the main concerns are allergenic proteins. It is of great importance to the whole manuscript.
Response: Thanks for your consideration. According to the reviewer’s suggestion, the sentence was adding. The biosensor is a device capable of detecting different compounds such as environmental pollutants [21] [22], the food additives [23], pesticide residue [24], [25], and biomolecules in biological [26] or food [27] samples (Lines 98-101).

5. What’s the advantage and functions of using nanoparticles in detecting allergenic proteins? There is necessarity to state the main role of nanoparticles and the relationship of them in a separate part.
Response: Thanks for your consideration. According to the reviewer’s suggestion, the sentence was added. The application of nanoparticles in the determination of allergenic proteins in foods, in general, brought as benefits the ability to obtain the limit of detection as shown in table 3. Thus, it was observed that the analytical methods that use AuNPs were the ones with the best limit of detection. (lines 504-507).

6. Is it possible to point out the future research direction of preparing biosensors to detect allergic proteins?
Response: Thanks for your consideration. According to the reviewer’s suggestion, the sentence was added. Associated with this, we can consider as perspective, the use of different component transduction systems together, such as optoelectric systems and the use of Nano-sized Molecularly Imprinted Polymers as Artificial Antibodies and hydrogel nanoparticles for the detection of allergenic proteins in foods. (lines 534-537).

7. The organizational structure of the article is a bit of chaotic, and the thinking is not clear enough. It needs to be well discussed for more clear. The classification should base on the methods or the objects or the nanomaterials. After selecting a category, other examples and narratives are interspersed, rather than listing various classifications and repeated station.
Response: Thanks for your consideration. According to the reviewer’s suggestion, the sentences were rewritten. For a better understanding, we separate the classic methods from the methods using biosensors. Allergen detection (line 235) and Biosensor technology (line 257).
8. Some refs about food samples detection should be added, such as
10.1016/j.foodchem.2020.127785, 10.1016/j.foodchem.2019.04.034,10.1016/j.foodchem.2020.127987, 10.1016/j.aca.2014.10.016, 10.1016/j.foodchem.2020.127974,10.1021/jf052489r.
Response: Thanks for your consideration. According to the reviewer’s suggestion, the references added were 10.1016/j.aca.2014.10.016 (reference 22) and 10.1016/j.foodchem.2020.127974 (reference 24).

We would like to thank the reviewers for his insight and attentive criticism of our Manuscript. We believe that we have responded to comments, thus increasing the quality of the manuscript.

Reviewer 2 Report

Authors overview here nanostructured biosensors for the detection of food allergenic proteins.

Although the topics discussed in this review, both Food Allergy and Nanostructured Biosensors, are of maximum relevance and the authors have reviewed a large number of publications this review attempts to cover too many fields but does not discuss in detail the information reported to make it instructive. In addition, the English and the organization should be improved and should be more careful with the selected terminology to flow well and be more attractive. I do not think that the information and approaches highlighted are sufficient for a review article, as the information collected is not discussed in detail and some more relevant and representative approaches should be included. I also do not think that the illustrations and tables are sufficient either in number or quality.

Therefore, I consider that should be reviewed and significantly improved:

-In terms of organization: section 3.2 should be established a clear distinction between conventional methodologies and biosensors discussing extensively their advantages and subsequently introducing the different classes of biosensors based on the type of transducer or the recognition bio-element (immuno-, geno- or apta-sensor); the title of the section 3.4.1 refers exclusively to AuNPs, but also other types of NPs are discussed; and section 3.4.5 should be introduced before the discussion of the different types of nanomaterials.

- Carefully review the terminology: nanoparticles or nanomaterials; line 306: element immunosensors; line 439: QDs or Qdots; line 462: “The immobilization of biomaterials on the surface, whether in the biosensor or nanoparticle” the term biosensor must be modified by transducer since a biosensor has an integrated receptor and a transducer device.

-The quality of the figures should be improved. Correct the axis titles in figure 3 (x: allergenic foods; y: number of allergenic proteins). Other figures should be included to show the reader the different nanostructures used in the biosensors fabrication to support sections 3.4.1-3.4.4.

- Tables should be included that summarize the most significant information highlighted in this review (allergen, detection technique, type of bioassay, immobilization strategy, nanomaterials used, analysis time, sensitivity...) of all the papers mentioned to support sections 3.4.1-3.4.4.

-This sentence should be carefully discussed and justified: “Unlike ELISA methods, where the antibody is used only once, in biosensors, the antibodies can be used several times [42], [54]”.

- The perspectives of bio-sensing for food safety should be discussed in detail.

Unfortunately, I consider that this review neither arouses the interest of the reader nor is of sufficient quality to recommend its publication in this Journal.

Author Response

Reviewer 2
1. In addition, the English and the organization should be improved and should be more careful with the selected terminology to flow well and be more attractive. I do not think that the information and approaches highlighted are sufficient for a review article, as the information collected is not discussed in detail and some more relevant and representative approaches should be included. I also do not think that the illustrations and tables are sufficient either in number or quality.
Response: Thanks for your concern. According to the suggestion the language was checked by a native English speaker and sentences were rewritten and others added. (lines 120-121, 177-181, 200,201, 206-209, 253-255, 259,260, 264-270,296-299,304-307,437-442, 450-453, 474-479, 489-493, 518-520). More Illustration was added and further information in the table. (lines 908, 956, 989).
2. In terms of organization: section 3.2 should be established a clear distinction between conventional methodologies and biosensors discussing extensively their advantages and subsequently introducing the different classes of biosensors based on the type of transducer or the recognition bio-element (immuno-, geno- or apta-sensor); the title of the section
Response: Thanks for your observation. We separated the section in conventional methodologies and biosensors as suggested by reviewer. (lines 235 and 257).
3- 3.4.1 refers exclusively to AuNPs, but also other types of NPs are discussed; and section
Response: Thanks for your observation. We added one sentence to explain other nanoparticles. Gold nanoparticles are widely used in combination with other nanoparticles to improve the detection system; for example, gold particles combined with quartz crystal microbalances provide excellent results when combined with biosensors. (lines 433-435).
4. 3.4.5 should be introduced before the discussion of the different types of nanomaterials.
Response: Thanks for your observation. According to the reviewer’s suggestion. We change 3.4.5 to 3.5.1 (line 372).

5. Carefully review the terminology: nanoparticles or nanomaterials;
Response: Thanks for your observation. According to the reviewer’s suggestion. We change nanomaterials to nanoparticles (line 340, 344, 346, 349, 413).
6. line 306: element immunosensors;
Response: Thanks for your observation. According to the reviewer’s suggestion. This sentence “dot” was deleted. Among the biological recognition elements (immunosensors) have been more (line 319).
7. line 439: QDs or Qdots;
Response: Thanks for your observation. According to the reviewer’s suggestion, the sentence was changed to QDs.
8. line 462: “The immobilization of biomaterials on the surface, whether in the biosensor or nanoparticle” the term biosensor must be modified by transducer since a biosensor has an integrated receptor and a transducer device.
Response: Thanks for your consideration. According to the reviewer’s suggestion, the sentence was changed to a transducer. (line 392).
9. The quality of the figures should be improved. Correct the axis titles in figure 3 (x: Allergenic foods; y: Number of allergenic proteins). Other figures should be included to show the reader the different nanostructures used in the biosensors fabrication to support sections 3.4.1-3.4.4.
Response: Thanks for your observation. According to the reviewer’s suggestion, the quality of the figures was improved (lines 896, 908, 956).
10. Tables should be included that summarize the most significant information highlighted in this review (allergen, detection technique, type of bioassay, immobilization strategy, nanomaterials used, analysis time, sensitivity...) of all the papers mentioned to support sections 3.4.1-3.4.4.
Response: Thanks for your observation. According to the reviewer’s suggestion, the essentials information were added in the table (line 989).
11. -This sentence should be carefully discussed and justified: “Unlike ELISA methods, where the antibody is used only once, in biosensors, the antibodies can be used several times [42], [54]”.
Response: Thanks for your observation. According to the reviewer’s suggestion, the sentence was revised (lines 289, 290).

12. The perspectives of bio-sensing for food safety should be discussed in detail.
Response: Thanks for your consideration. According to the reviewer’s suggestion, the sentences were added. Associated with this, we can consider as perspective, the use of different component transduction systems together, such as optoelectric systems and the use of Nano-sized Molecularly Imprinted Polymers as Artificial Antibodies and hydrogel nanoparticles for the detection of allergenic proteins in foods. (lines 536-539).

We would like to thank the reviewers for his insight and attentive criticism of our Manuscript. We believe that we have responded to comments, thus increasing the quality of the manuscript.

Reviewer 3 Report

The authors have captured the essence, importance, and advances in the field of Biosensors for food allergens in a concise and precise manner.

The review article is acceptable in the present form although a few suggestions are as below

1. The authors have presented the methodology for the selection of articles discussed in this review very clearly and precisely. Although Big Eight allergens are quite popular, please elaborate for readers with minimal expertise in the allergens field in the introduction. Description of the Big Eight was not given until Section 3.1

2. Focus Questions, a comment on detection limits from various techniques would be very useful

3. Incorporate different techniques currently available for allergen detection in a tabular form

4. Comment on how these technologies facilitate and/help in advances for POC devices

Author Response

Reviewer 3
1. The authors have presented the methodology for the selection of articles discussed in this review very clearly and precisely. Although Big Eight allergens are quite popular, please elaborate for readers with minimal expertise in the allergens field in the introduction. Description of the Big Eight was not given until Section 3.1.
Response: Thanks for your consideration. According to the reviewer’s suggestion, the sentence was adding. According to a report published by the Food and Agriculture Organization (FAO) of the United Nations, the most critical allergenic foods are soybean, cow's milk, chicken eggs, peanuts, crustaceans, tree nuts, cereal sources of gluten and fish the called Big Eight. (lines 77-80).
2. Focus Questions, a comment on detection limits from various techniques would be very useful
Response: Thanks for your consideration. According to the reviewer’s suggestion, the sentences were added. The application of nanoparticles in the determination of allergenic proteins in foods, in general, brought as benefits the ability to obtain limit of the detection as shown in table 3. Thus, it was observed that the analytical methods that use AuNPs were the ones with the best limit of detection. (line 505-508).
3. Incorporate different techniques currently available for allergen detection in a tabular form
Response: Thanks for your consideration. According to the reviewer’s suggestion, the table was added (line 975).
4. Comment on how these technologies facilitate and/help in advances for POC devices
Response: Thanks for your consideration. According to the reviewer’s suggestion, the sentence was added. (line 92-94).

We would like to thank the reviewers for his insight and attentive criticism of our Manuscript. We believe that we have responded to comments, thus increasing the quality of the manuscript.

Reviewer 4 Report

The manuscript, titled "A systematic review of Food Allergy: Nanobiosensor and allergenic protein detection” shows a systematic review article organized by the category of allergen and nanoparticle detection.

I lack experience in the methodologies used for the selection of the articles used in this systematic review. I miss relevant papers in the allergen detection has not found using these methodologies, even reviews published in this journal a short time ago.

Therefore, as this is the objective of the work, I believe that the result has not been as expected.

Author Response

We would like to thank the reviewers for his insight and attentive criticism of our Manuscript. We believe that we have responded to comments, thus increasing the quality of the manuscript.

Round 2

Reviewer 1 Report

It was well revised and can be accepted.

Author Response

(The authors gave the same response as above.)

Reviewer 2 Report

Authors overview here nanostructured biosensors for the detection of food allergenic proteins. The authors have made an exhaustive revision of the paper after the first revision, improving significantly the organization and content and therefore improving the reading flow of the paper, according to the relevance of the topic presented as well as the mentioned papers. Therefore, I consider it deserves publication in this Journal after addressing the following minor concerns:

- In figure 2 the immobilized antibodies on nanoparticles should not be considered as part of the sample matrix, since they are added externally for the biosensor fabrication.

-The figure 5 caption cannot refer to the immobilization of bioreceptors since it does not specifically represent any type of immobilization or protocols or steps for the bioreceptor immobilization. This figure should be improved by representing in a schematic manner the different types of immobilizations described in section 3.5.1.

-line 482: correct body by antibody.

This is a review with enough novelty, quality and relevance to deserve its publication in this Journal after addressing the minor issues pointed out.

Author Response

  1. In figure 2 the immobilized antibodies on nanoparticles should not be considered as part of the sample matrix, since they are added externally for the biosensor fabrication.

Response: Thanks for your observation. According to the reviewer’s suggestion, we removed immobilized antibodies on nanoparticles in figure 2 (line 111).

  1. The figure 5 caption cannot refer to the immobilization of bioreceptors since it does not specifically represent any type of immobilization or protocols or steps for the bioreceptor immobilization. This figure should be improved by representing in a schematic manner the different types of immobilizations described in section 3.5.1.

Response: Thanks for your observation. Figure 5 was improved as suggested by the reviewer. (Line 355).

  1. line 482: correct body by antibody.

Response: Thanks for your observation. According to the reviewer’s suggestion, we changed body by antibody. (Line 488).

We would like to thank the reviewers for his insight and attentive criticism of our Manuscript. We believe that we have responded to comments, thus increasing the quality of the manuscript.

Reviewer 4 Report

I miss relevant papers in the allergen detection on the other hand the tables should summarized the most significant information of all the papers mentioned, for instance Lane.262-266...... Furthermore, tools  that are based on the detection of nucleotide sequences such as DNA or RNA of allergenic proteins are used too, such as polymerase chain reaction (PCR) and its variants such as multiplex-real-time PCR, PCR-ELISA, duplex PCR [54], [10], [55] (Table 2) But the table 2 not includes any reference about the PCR application on the allergen detection

In the review there is no reference to the work of the Pingarron group some included in the Special Issue "Biosensors Applications for the Detection of Food Contaminants and Adulterants"

Author Response

  1. I miss relevant papers in the allergen detection on the other hand the tables should summarized the most significant information of all the papers mentioned, for instance Lane.262-266...... Furthermore, tools  that are based on the detection of nucleotide sequences such as DNA or RNA of allergenic proteins are used too, such as polymerase chain reaction (PCR) and its variants such as multiplex-real-time PCR, PCR-ELISA, duplex PCR [54], [10], [55] (Table 2) But the table 2 not includes any reference about the PCR application on the allergen detection.

Response: Thanks for your observation. According to the reviewer’s suggestion, we added more references in table 2 and in the text about DNA detection. (line 72, 272,776).

  1. In the review there is no reference to the work of the Pingarron group some included in the Special Issue "Biosensors Applications for the Detection of Food Contaminants and Adulterants"

Response: Thanks for your consideration. According to the reviewer’s suggestion, we added more references. (Lines 809,824).

We would like to thank the reviewers for his insight and attentive criticism of our Manuscript. We believe that we have responded to comments, thus increasing the quality of the manuscript.
